# A pilot study on bio-banding in male youth ice hockey: Players' perceptions and coaches' selection preferences

Oliver Lindholm[1], Erik Niklasson[1,2], John Lind[3], Daniele A. Cardinale[4,5], Tommy R. Lundberg[1,6]*

1 Division of Clinical Physiology, Department of Laboratory Medicine, Karolinska Institutet, Stockholm, Sweden, 2 Centre for Physical Activity Research, Copenhagen University Hospital—Rigshospitalet, Copenhagen, Denmark, 3 Swedish Ice Hockey Association, Stockholm, Sweden, 4 Department of Physiology, Nutrition and Biomechanics, The Swedish School of Sport and Health Sciences (GIH), Stockholm, Sweden, 5 The Swedish Sports Confederation (Riksidrottsförbundet), Stockholm, Sweden, 6 Unit of Clinical Physiology, Karolinska University Hospital, Stockholm, Sweden

* tommy.lundberg@ki.se

**Data Availability Statement:** The raw data are available in Supporting file S2. To comply with the ethical approval and avoid the risk of individual

## Abstract

Classifying athletes based on estimates of biological maturation (i.e., bio-banding) as a supplement to traditional age grouping has been shown to be a potential tool for enriching player development in team sports; however, bio-banding has not yet been evaluated in ice hockey. The primary aim was to investigate player experiences and coaches' selection preferences in bio-banding versus age-banding in a group of 12-13-year-old (early growth spurt) male elite players (n = 69). We also examined the relationship between somatic maturity, expressed as a % predicted adult height (%PAH), and fitness performance. Bio-banding was assessed using a questionnaire and 29 coaches selected their top players in each game based on age or bio-bands. %PAH correlated with grip strength (r = .57, p>0.001) and jumping power (r = .63, p<0.001), but not with vertical jump height, sprint time or endurance. Players who played against more mature players in bio-bands than in age groups experienced higher demands, while players who played against less mature players were able to utilize their skills to a greater extent. Coaches generally favored later-than-average maturing players who performed better on performance tests and chronologically older players in bio-banding. We conclude that bio-banding in youth ice hockey has some promising effects and warrants further evaluation.

## Introduction

Biological maturation has a major impact on the physical performance of adolescent male athletes. Apart from stature and mass, previous studies have shown a relationship between advanced maturity and increased countermovement jump performance, agility, sprint speed, endurance, and intermittent team sport performance [1–3]. During adolescence, somatic maturity can vary by up to 5 years within a birth year [4]. Thus, athletes of the same chronological age may have significant differences in size and athleticism and therefore require different challenges to develop optimally during training and competition at the elite junior level.

players being identified by the combination of different variables, date of birth, height and weight have been excluded. Requests for this data can be directed to the corresponding author. Non-author contact information for data requests: Thomas Gustafsson Thomas.gustafsson@ki.se (head of division of Clinical Physiology, Karolinska Institutet). Long-term data storage and availability is guaranteed by the guidelines for research documentation under Swedish law and the guidelines of the corresponding author's academic institution.

**Funding:** The author(s) received no specific funding for this work.

**Competing interests:** TRL has received financial compensation from the Swedish Ice Hockey Association for consultancy work. JL is employed by the Swedish Ice Hockey Association. EN has received reimbursement of travel expenses from the Swedish Ice Hockey Association. The specific roles of these authors are articulated in the 'author contributions' section. The Swedish Ice Hockey Association did not have any additional role in the study design, data collection and analysis, decision to publish, or preparation of the manuscript. This does not alter our adherence to PLOS ONE policies on sharing data and materials (see data availability statement).

In line with the benefits of advanced maturation, youth soccer academies tend to disproportionately select early-maturing players (maturity bias), suggesting a tendency among talent scouts to equate physical maturity with future potential [5, 6]. To address this issue and create equal development opportunities, attempts have been made to categorize soccer players by maturity as a complement to chronological age groups [7]. This strategy, known as bio-banding, has shown great promise as it reduces the physical demands on less mature players and allows more mature players to play a more physically, technically, and tactically challenging game without increasing the risk of injury [7–9]. Although the concept of bio-banding has shown promise in Premier League soccer academies [10], it is yet to be explored in ice hockey.

Ice hockey is physically demanding and requires anaerobic power, frequent intermittent acceleration and deceleration; games involve frequent physical contact and collisions [11]. As biological maturation is associated with an increase in muscle mass, anaerobic power and oxygen uptake, early maturation may be beneficial in physical sports such as ice hockey [12, 13]. Indeed, it has been reported that 14 to 16-year old male players selected in ice hockey try-outs are more mature, physically larger and more athletic than unselected players [14–16], which is consistent with previous research in other contact sports such as soccer and rugby [5, 6, 17]. Furthermore, a recent retrospective register study based on data from Swedish high school ice hockey academies showed that the U16 and U18 national teams consisted of more early maturing players, while the U20 national team had an even distribution between early, on-time and late maturing players [18]. Thus, a selection bias towards early maturation seems evident in youth ice hockey. This provides a compelling case for bio-banding in ice hockey to complement chronological age-bands, as it can challenge this bias by reducing the physical differences between players and providing players with new challenges that promote long-term athletic development.

Bio-banding would also provide the opportunity to evaluate players in a different format, which could affect coaches' selection preferences. Some evidence from soccer suggests that when coaches know the maturity of players (through shirt numbering), they are more likely to select late-maturing players [19]. However, it is not known whether bio-banding can reduce maturity-related selection biases in team sports, and it would be particularly important to investigate this in a physical sport such as ice hockey.

Maturity-related variation tends to intensify around the pubertal growth spurt and before peak height velocity, typically before the age of 14 in males, and can influence team composition and talent identification. In this pilot study with U13 and U14 teams of an elite Swedish youth ice hockey club, our primary aim was therefore to evaluate players' perceptions and coaches' selection preferences in bio-banding compared to age-banding. In addition, the study investigated the potential overrepresentation of early maturing players, and examined whether maturity in these age groups correlates positively with physical performance.

## Methods

### Study design and population

The study was conducted on three different days within 11 days. The first study day consisted of anthropometric measurements, off-ice physical fitness testing, and documentation of the biological parents' self-reported heights. The second study day consisted of games in age groups and on-ice sprint tests. The third day consisted of games in bio-bands and an evaluation of the players' perceptions using questionnaires. The games were recorded on video and 29 coaches from elite organizations were asked to select 7 players from each game (described below).

From the academy of a Swedish elite ice hockey club, 41 males and 2 females in the under 13 team (U13) and 40 males and 1 female in the under 14 team (U14) were invited to participate in the study. Of these, 36 males and 2 females from U13 and 33 males and 1 female from U14 participated. One player was excluded due to being too old (i.e., by definition not a part of the under 14 team). While the females were allowed to participate in the games, they were excluded from the analysis due to large sex differences in maturation [20].

All players, parents, and coaches received written and verbal information about the study. They had time to ask questions and were informed that they could withdraw their consent for participation at any time without consequences for their regular team activities. A written informed consent form was signed by both the players and their legal guardians if they chose to participate. Ethical approval was granted by the Swedish Ethical Review Authority, reference number 2022-02457-01.

### Biological maturation assessment

The Khamis-Roche method was used to estimate maturity, expressed as percentage of predicted adult height (%PAH), due to its reliability and non-invasive ease of use in the field [6]. Height was measured to the nearest 0.1 cm using a roll-up tape measure with a wall attachment (Seca, seca 206, Germany). Body mass was measured using a digital scale with an accuracy of 0.1 kg (Beurer GmbH, BG 19, Germany) and age was expressed as decimal age on the day of physical performance testing. Parents' self-reported heights were recorded on the day of testing and adjusted for overestimates [21]. Using sex-specific smoothed intercepts and coefficient values from the Fels Longitudinal Study data, the measures were then substituted into the equation: $\beta_1$ *stature* + $\beta_2$ *weight* + $\beta_3$ *mid–parent stature* [22]. Consistent with previous research, categorical ranges for maturity status were established and defined as pre-PHV < 88% PAH, circa PHV 88–96% PAH, and post-PHV > 96% PAH [23]. Furthermore, maturity timing was calculated as standard deviation Z-scores using age- and sex-specific means and standard deviations from a reference population of 3650 healthy Swedish children [20]. Absolute height values were recalculated as %PAH with height at age 18 years as adult height. Thresholds for maturity timing were set at ± 0.5 (i.e., on-time = Z-score 0.5 to -0.5, early = Z-score > 0.5, and late = Z-score < -0.5), as this statistically results in approximately three equally sized groups in the general population [24].

### Fitness performance

The performance of the players in the squat jump (SJ) without arm swing was performed on a contact mat (SMARTJUMP, FusionSport, Australia) [25]. Each player was allowed three attempts and the best jump hight result was recorded in cm, as well as peak power (W) and peak power relative to body mass (W·kg⁻¹). Hand grip strength (HGS) was measured using a digital hand grip dynamometer (T.K.K 5101 Grip d, Takei, Japan) [26]. When performing the HGS test, players stood up and started with their hand above their heads, fully extending their elbows. They pressed the dynamometer with maximum force for 5 seconds, performing a 180-degree movement in the shoulder with an extended elbow. The angle of the wrist was determined by the participant [27]. Each player was allowed two trials with each hand, and the best score was recorded in kilograms of force. Endurance performance was measured using the YoYo Intermittent Recovery Test Level 1 (YYIRT), which consists of short sprints at an increasing pace with 10 seconds of rest between sprints, repeated until failure [28, 29]. The result of YYIRT was recorded as the number of meters run at completion. If the player did not reach the marked distance in time, a warning was given, and on the second failure, the player was considered to have reached failure. On-ice performance was measured using 30-meter

sprints and recorded using photocells (SmartSpeed, Fusion Sport, United Kingdom). The player started 0.5 meters behind the first timing gate and stood still. Each player was allowed 2 attempts. Goalkeepers were excluded from the on-ice sprint test.

## Bio-banding

The first day of intervention consisted of age-banded games (AB, U13 vs. U13 and U14 vs. U14). On the second day, players were divided into two bio-bands with comparable number of players, Bio-Band Lower (BBL) ≤ 87.4% PAH and Bio-Band Higher (BBH) ≥87.5% PAH. Each group was then divided into two opposing teams set by team coaches with an ambition to make them as even as possible. The games were played on an Olympic size ice rink over two 20-minute periods with 5 players and a goalkeeper on the ice. Player positions were documented.

## Player perceptions

After participating in both formats, players evaluated bio-banding compared to age-banded games by completing a questionnaire containing statements and an adjacent 5-point Likert scale: "strongly disagree", "disagree", "neither agree nor disagree", "agree" or "strongly agree", similar to previously used questionnaires in soccer [7]. The questionnaire was designed to assess players' perceptions on four subscales: social, psychological, technical/tactical, and physical. For the analysis of this questionnaire, the players were divided into three groups: players who played against more mature opponents than usual (MMO players), i.e. players moving from age-band U13 to BBH, and players who played against less mature opponents than usual (LMO players), i.e. age-band U14 who played in BBL, and players who played as usual against opponents of approximately equal maturity (EMO players), i.e. age-band U13 and BBL or age-band U14 and BBH.

## Coaches' selection preferences

The 29 coaches who were asked to assess and select players came from either an elite club in Sweden or the regional offices of the Swedish Ice Hockey Association. They had a mean age of 43 years. All coaches had extensive experience in the sport with an average of 15 years as a coach or scout and 23 coaches had taken one of the two highest-level courses within the coach education system provided by the Swedish Ice Hockey Association. Each coach was randomly assigned to evaluate either the two age-banded games or the two bio-banded games. For each game, they were asked: *"Which 7 players would you pick from this game if you were in charge of admissions to an ice hockey high school team (or equivalent)?"*. No directions were given regarding number of picks per team or position. In order to properly assess players in both teams, the coaches were asked to watch each game twice.

## Statistical analyses

The data were normally distributed according to visual inspection of histograms. Mean values were used to describe maturity timing for age-bands and positions. Means were tested against the null hypothesis using a one-sample t-test compared to the reference population and a two-sample t-test between bands and between positions. Pearson's correlation was used to analyze the relationship between maturity level, decimal age, and physical performance. Mean scores, ANOVA and t-tests were used to describe the general maturity status, physical performance, and anthropometric data of groups (U13, U14, and AB, BB). The Likert scale data were analyzed in a descriptive manner with density plots, but also with marked mean values, following

Towlson et al. [30], in order to visualize the general opinion of the group of players, taking into account the variance [31]. Means for each coach's selected players were calculated. For each variable, an overall mean was calculated for the coaches evaluating BB and AB, respectively. An average player was calculated from the mean of all players who participated in the games. The overall means of the coaches were tested with a t-test (two samples) against each other and with a t-test (one sample) against the average player. The relationship between the number of selections from coaches and various other metrics was explored using Spearman's rank correlation. The odds ratios of player selection were calculated for offensive (forwards and centers) and defensive players (defencemen and goalkeepers). A p-value of $< .05$ was considered significant. Statistical tests and figures were performed and generated in R 4.2.1 [32] and Rstudio 2022.09.23 [33].

# Results

## Player characteristics

General player chareteristics are presented in Table 1. The mean decimal age for the entire group was 12.8 years (range: 11.7–13.7 years). The two age-bands differed significantly in % PAH (Table 1 and Fig 1). The vast majority of players were considered on-time in their maturity timing, with fewer being categorized as Early and fewest as Late (Table 1). The mean Z-score for maturity timing was not different from the reference population (p = .13). As for positions, defencemen had an earlier maturity timing than the reference population (mean difference 0.23, 95%CI 0.027–0.44, p = .029) while goalkeepers and offensive players (forwards and centers) did not differ compared to the reference population. Defencemen and goalkeepers were both significantly earlier in timing than offensive players (mean difference: 0.29, p = .021 and 0.4, p = .026, respectively) while no difference was observed between defencemen and goalkeepers (Table 1).

HGS and SJ peak power correlated more strongly with %PAH than with decimal age, while relative SJ peak power (W·kg$^{-1}$) was moderately correlated with %PAH (Fig 2). Neither SJ height, YYIRT distance, nor on-ice 30m sprint time correlated with %PAH nor with decimal age.

Compared to AB, BB decreased within-game mean ranges while increasing between-game differences in parameters such as %PAH, Z-score, weight, height, HGS and SJ peak power, while the opposite was found for decimal age (Fig 3).

**Table 1. General descriptives of players.**

| | U13 (n = 36) | U14 (n = 33) | All (n = 69) |
|---|---|---|---|
| Decimal age (years) | 12.3 (0.3) | 13.3 (0.3) | 12.8 (0.6) |
| %PAH | 85.9 (2.1) | 89.0 (2.1) | 87.4 (2.6) |
| Maturity timing Z-score | 0.13 (0.5) | 0.04 (0.45) | 0.09 (0.47) |
| *Goalkeepers* | 0.03 (0.36) | 0.66 (0.39) | 0.35 (0.48) |
| *Defenders* | 0.29 (0.61) | 0.19 (0.30) | 0.23 (0.44) |
| *Offensive players* | 0.07 (0.48) | -0.21 (0.36) | -0.06 (0.44) |
| Maturity timing classification (%) | | | |
| *Early* | 22% | 15% | 19% |
| *On-time* | 69% | 76% | 73% |
| *Late* | 8% | 9% | 9% |

Note: Data are mans and SD. Frequencies are reported as % only.

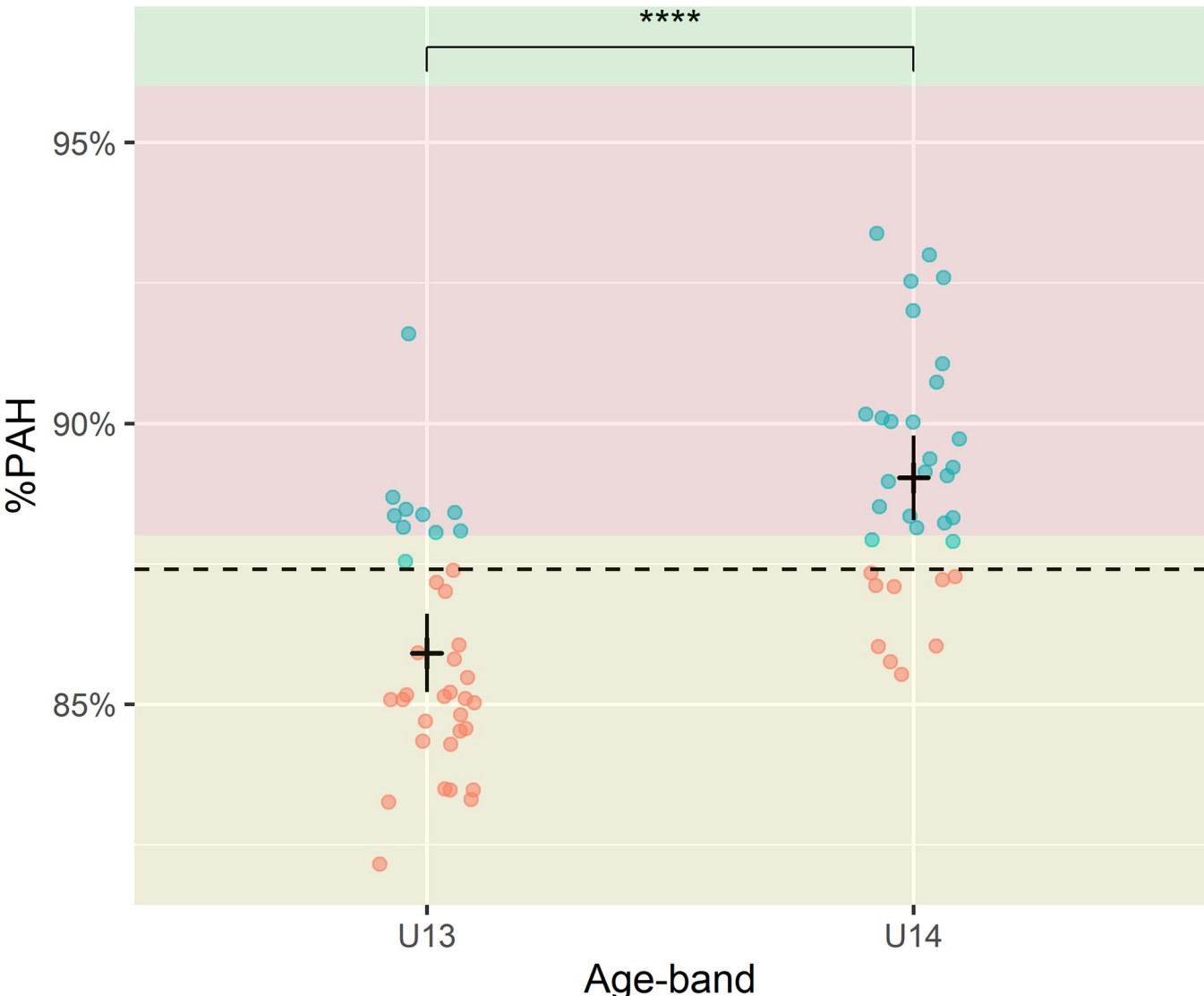

**Fig 1. Maturity status in age bands.** Symbol color: orange = BBL, blue = BBH. Zones for maturity status are indicated in different colors: yellow = pre-PHV, red = circa-PHV, green = post-PHV. The horizontal black line is the mean for respective age band and vertical black line is 95% CI. Dotted horizontal line is the grand mean of all players. %PAH = percentage of predicted adult height. U13 = Under 13 years, U14 = Under 14 years. ****: p ≤ 0.0001.

### Player perception in bio-bands

In general, MMO-players experienced a more physically, technically, and tactically challenging game, an increased need for concentration, increased demand to play well, and more effort was needed to succeed (Fig 4). They reported that they could use their strength and speed to their advantage. MMO-players were also more aware of their strength and weaknesses, had to release the puck earlier, and experienced more opportunities to try something new. LMO-players did not experience a more physically, technically, or tactically challenging game but instead, they had a more leading role in the team, could affect the game more, and could use speed to their advantage. They also experienced more opportunities to create or move into space, could travel with the puck for a longer duration of time, try something new, control the puck under pressure, and were more able to beat opponents to loose pucks. EMO-players

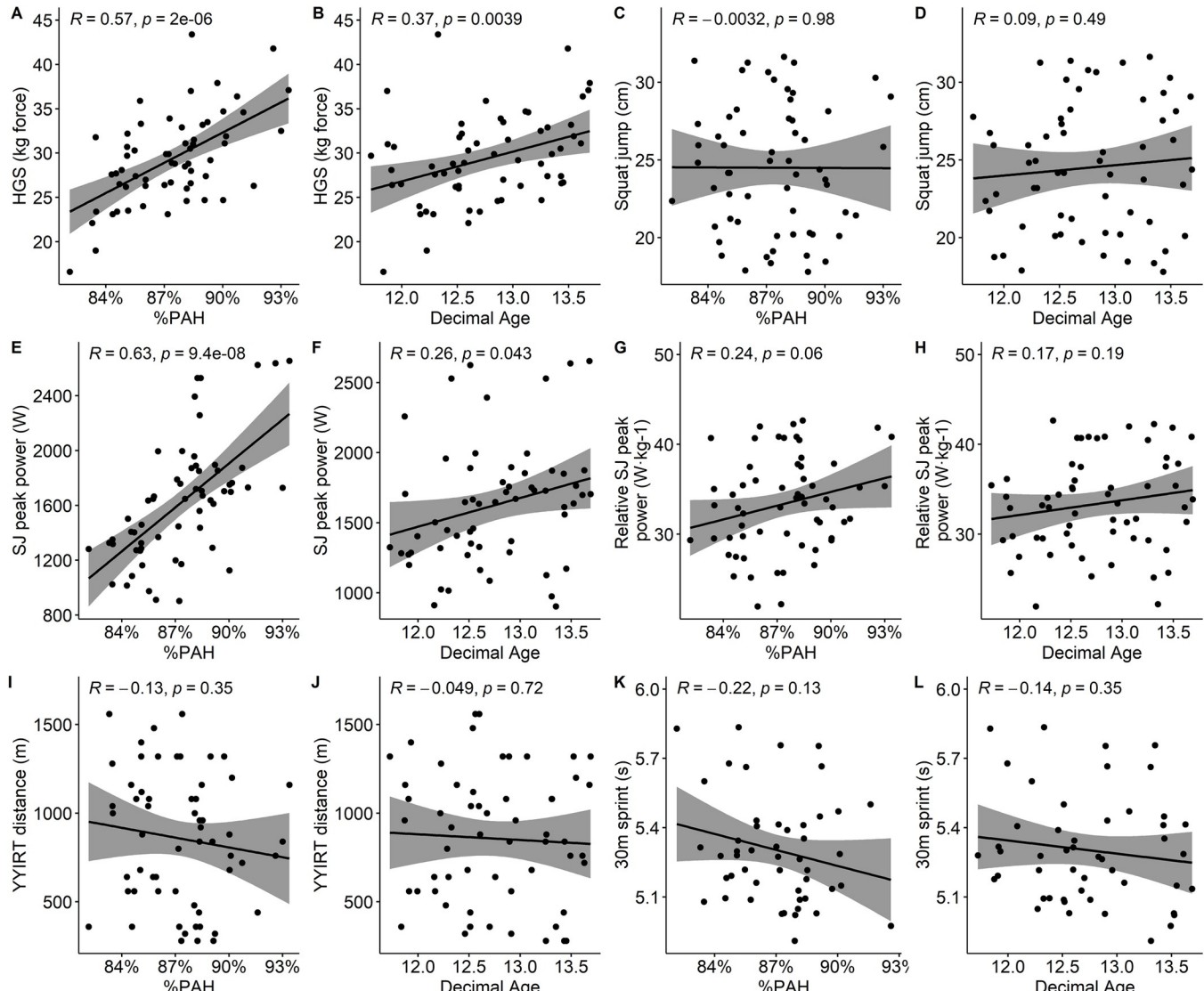

**Fig 2. Pearsons' correlation between physical performance tests and maturity status and chronological age.** %PAH: percentage of predicted adult height. HGS: Hand grips strength (kilogram force), SJ: Squat jump, YYIRT: YoYo Intermittent Recovery Test level 1.

experienced greater opportunities to use their technical abilities and speed to their advantage. All groups experienced an increased need to communicate and work as a part of a team, and they also experienced an opportunity to make new friends. No group perceived an increased risk of injury. A summary of Likert scale responses is presented in Fig 4. The results are presented in full in S1 File. Supporting raw data are available in S2 File.

## Coaches' selection preferences

Differences in player selection between formats were mainly observed in maturity status (% PAH), age, and on-ice speed. In AB, coaches preferred lower maturity status compared to bio-bands (mean difference: 0.58%PAH, p < .001). They also selected players with lower SJ peak power compared to the average participating player (Fig 5). In BB, coaches preferred chronologically older players (mean difference: 0.21 years, p < .001) and players with better results in

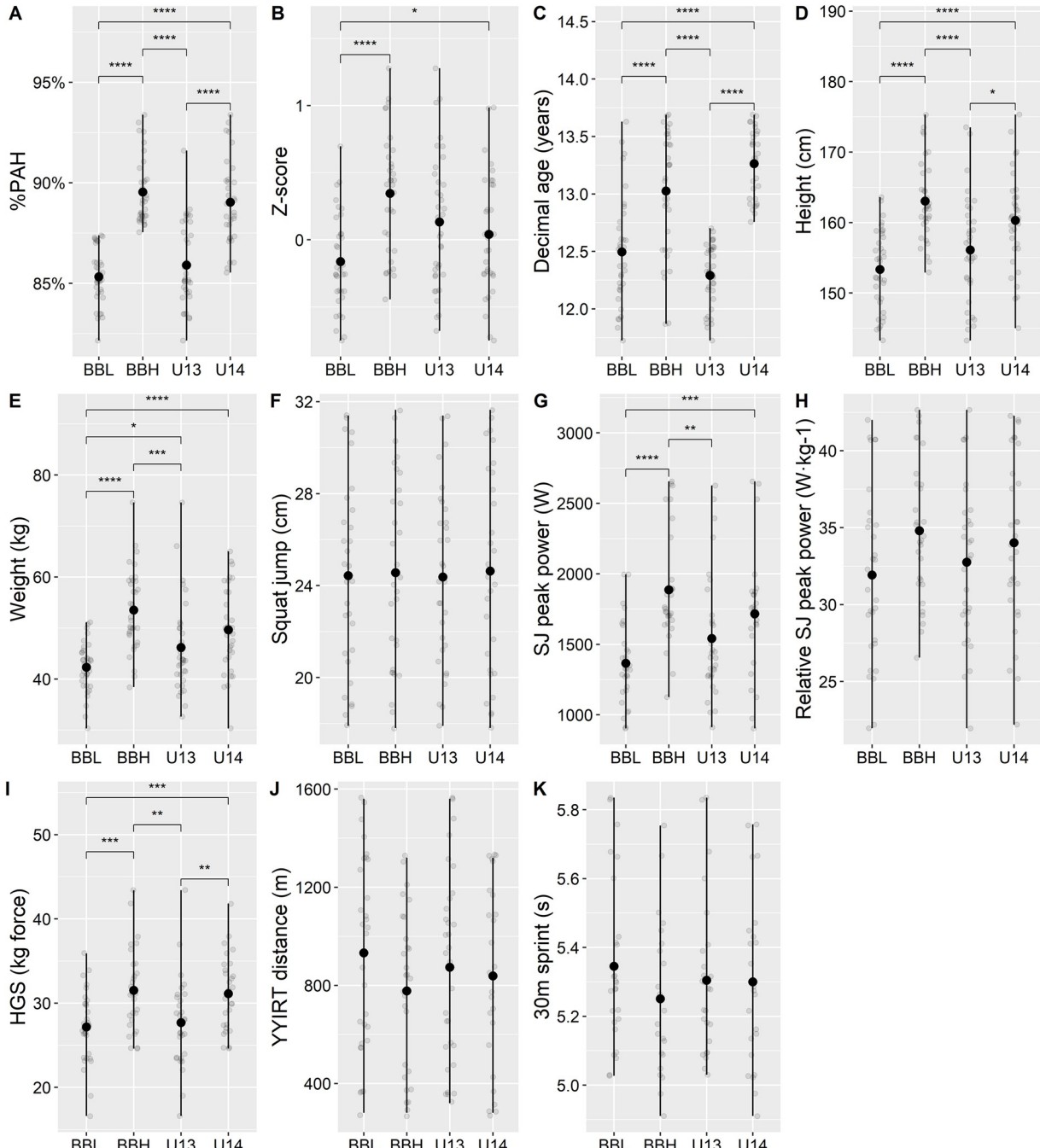

**Fig 3. Anthropometrics and physical performance in games.** BBL and BBH are bio-banded games (BBL = 82.2%–87.4% PAH, BBH = 87.5%–93.4% PAH), U13 and U14 are age-banded games (U13 = 11.7–12.7 years old, U14 = 12.8–13.7 years old). Black symbols mark the mean value. Black vertical lines: ranges within games. SJ = Squat Jump, HGS = Hand grip strength, YYIRT = Yoyo intermittent recovery test level 1. 30-meter sprints were performed on ice. *: p ≤ 0.05, **: p ≤ 0.01, ***: p ≤ 0.001, ****: p ≤ 0.0001.

the on-ice sprint test (mean difference: 0.04 sec, p = .031), compared to AB games. They also selected players with higher relative power output in the squat jump. Regardless of grouping format, coaches preferred players with a shorter predicted adult height, later maturity timing, smaller players in terms of weight and height, higher squat jump performance, higher force in HGS, faster time in 30-meter sprints on ice, and longer distances on the YYIRT-test (Fig 6 and

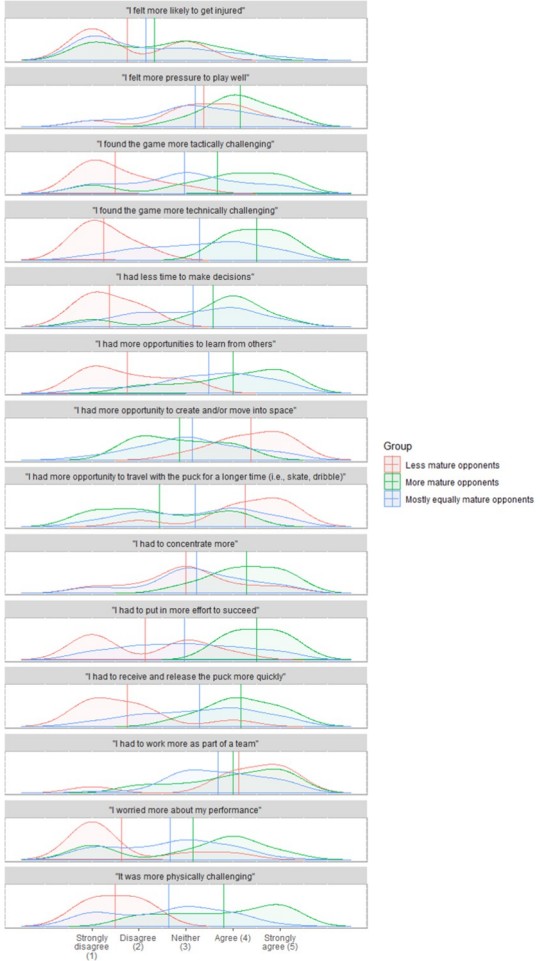

**Fig 4. Summary of player perception in bio-bands compared to age-bands.** Data are presented as Density plots from Likert-scale responses. Statements are shown above the respective density plot. Less mature opponents: players assigned to U14 and BBL, More mature opponents: players assigned to U13 and BBH, Mostly equally mature opponents: players assigned to U13 and BBL or U14 and BBH. Vertical line: mean for each group.

S1 File). The odds ratio for an offensive player being selected compared to a defensive player (i.e, defenceman or goalkeeper) was 1.99 in BB and 2.31 in AB.

## Discussion

This study aimed to investigate the suitability of bio-banding in adolescent ice hockey and the wider role of biological maturity in male players about to enter the accelerated phase of the growth spurt. While there was no significant maturity bias in this group of players compared to the reference population, the distribution of maturity timing between player positions was clearly skewed, with defenders maturing significantly earlier than offensive players. Maturity status correlated positively with absolute, but not relative, strength and power tests. As expected, the introduction of bio-banding reduced within-game variation in anthropometrics, maturity, and various fitness tests, while increasing the variation in chronological age. The majority of players found bio-banding to be a learning experience that offered new challenges and a format that emphasized teamwork. Players who played against more mature opponents than usual felt that they had to concentrate more, that it was more technically and tactically

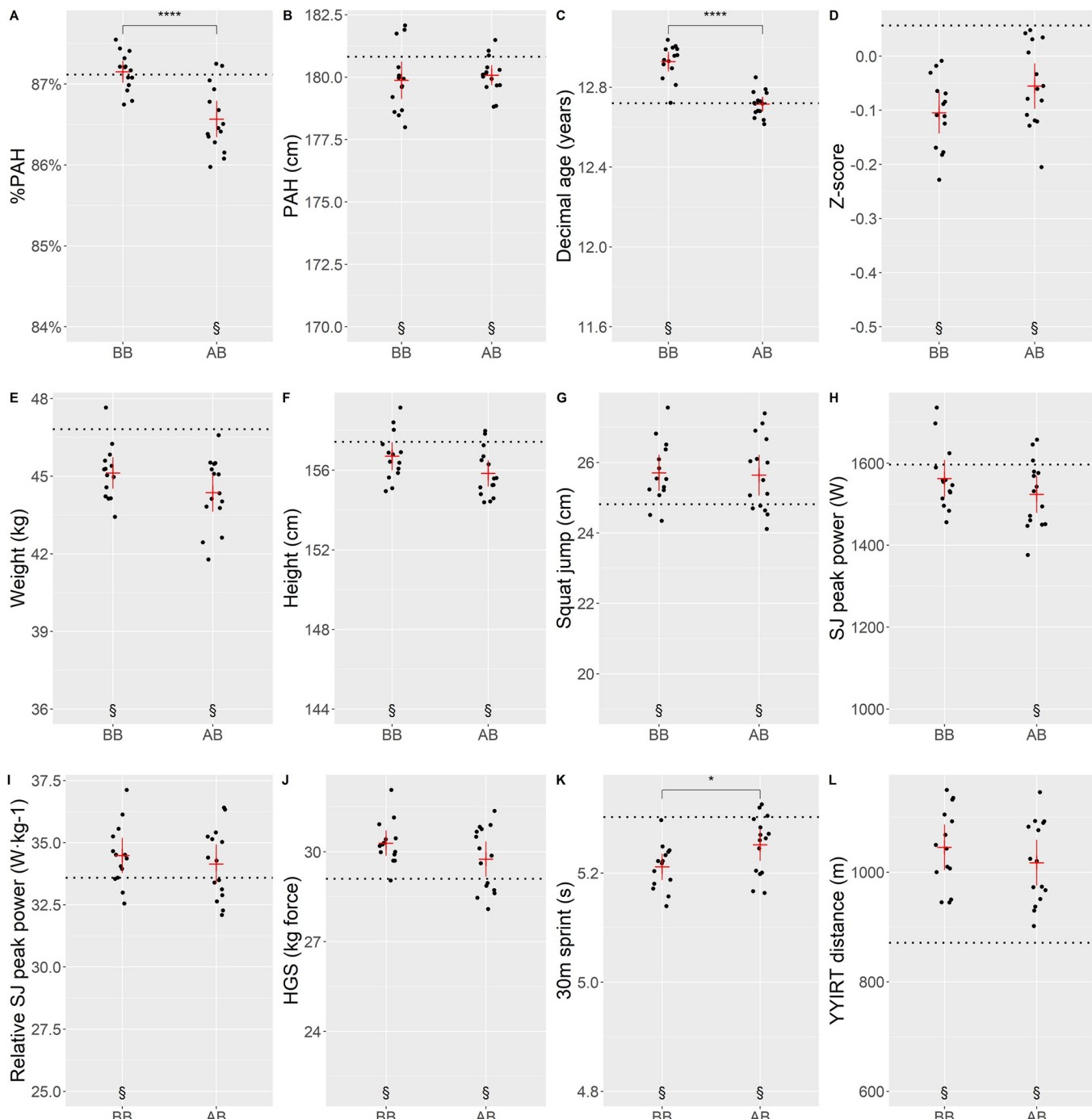

**Fig 5. Coaches' player selection in BB and AB.** %PAH: percentage of predicted adult height, PAH: predicted adult height, SJ: squat jump, YYIRT: YoYo Intermittent Recovery Test level 1. Each point is one coach's mean value of selected players. Horizontal red line: grand mean for coaches. The vertical red line indicates the 95% confidence interval of the grand mean. Black dotted horizontal line: average of all players. P-value of two-sample t-test comparing BB and AB is shown. §: $p \leq 0.05$ in one sample t-test for grand mean per format vs. the average player. *: $P \leq 0.05$ in two sample t-test between formats. ****: $P \leq 0.0001$ in two sample t-test between formats.

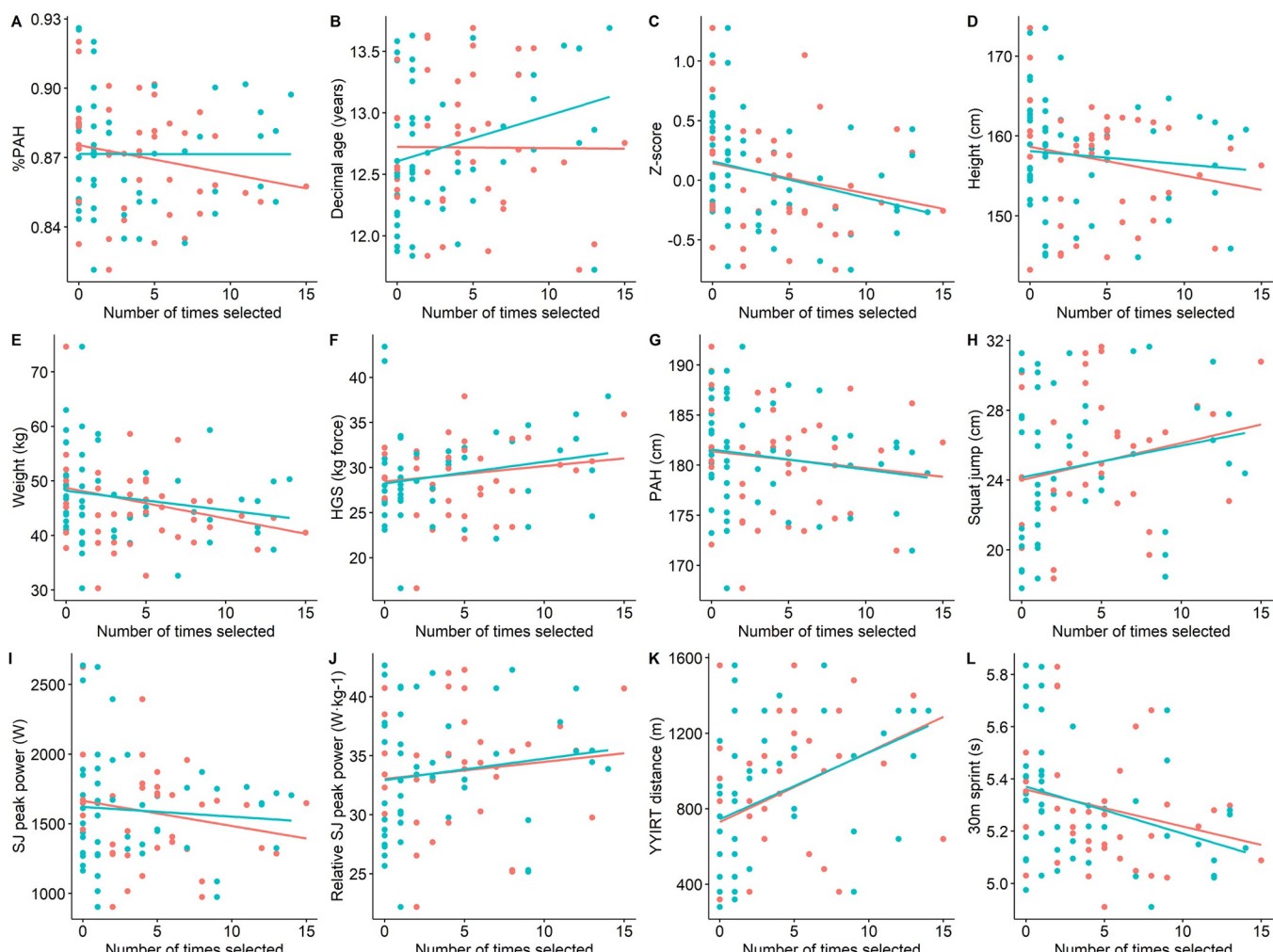

**Fig 6. Relationship between number of selections and various metrics of maturity, anthropometrics, and fitness performance.** %PAH: percentage of predicted adult height, PAH: predicted adult height, SJ: squat jump, YYIRT: YoYo Intermittent Recovery Test level 1. AB: age-bands, BB: bio-bands. See also related correlation data in S1 File.

challenging, and that they had to work harder to succeed. In contrast, players who played against less mature opponents than usual saw the opportunity to use their physique to their advantage and utilize their technical and tactical skills to a greater extent. When evaluating coaches' selection preferences, they generally favoured smaller, later maturing players who performed better on strength, endurance, and jumping tests. Coaches who evaluated players in the bio-banding format favoured chronologically older and faster players, while coaches who evaluated players in age-bands favoured smaller players.

## Maturity timing distribution

The uneven distribution of maturity timing between positions indicate that maturity plays an important role in players' competitive strategies already at this age and/or that coaches select players with a certain maturity for certain positions. The size and strength of early-maturing players may be more advantageous in defensive positions, where holding off opponents and winning physical duels is important, while later maturation seems to be more suitable for

forwards, perhaps aided by better agility. The tendency for early maturation in defensive positions as a phenomenon is supported by similar findings in soccer, where maturity biases have been observed in positions that typically require a larger body size [34].

## Players' perception of bio-banding

The bio-banding format was perceived to provide new challenges to the players when compared with the traditional age-bands, similar to previous findings in soccer, where early maturers have found bio-banding as more physically, tactically and technically challenging, offering them new opportunities for development, while late maturers have felt more competitive in technical, physical and psychological domains [7, 10]. The experience of players in bio-bands was further reinforced by the fact that coaches in this format favored older and later maturing players, suggesting a more prominent and dominant role for later maturing players in the team. While the short-term benefits of a more prominent role for late maturing players in bio-bands are obvious, it is important to recognize the positive effects of bio-banding for both early and late maturing players. Late maturing players may not perform as well on junior teams as their early maturing counterparts, but they have been reported to outperform early maturing players when it comes to succeeding on elite adult teams [18, 35]. From the perspective of early maturing players, bio-banding may seem disadvantageous at first glance, as they are deprived of the immediate performance recognition they would have received in age-based groups. However, in the long run, early maturing players may benefit the most from bio-banding. It encourages the development of strategies that are effective against similarly sized or larger players, which is valuable in competition with adults and crucial for technical development and experiencing continuous challenges during training and competition. Overall, our findings support the integration of bio-banding alongside age-banding in ice hockey to improve the overall development of players.

## Coaches' selection preferences

Surprisingly, coaches favoured smaller, later maturing players when it came to who they would select for a high school draft. This is in contrast to previous research in soccer where early maturing players are consistently favoured during adolescence [5, 6]. Despite careful wording in our communication with the coaches before the evaluation, we must consider the possibility that they may have perceived their talent recognition skills, particularly in distinguishing between talent and maturity, as under scrutiny. This may have led them to adjust their preferences and possibly overvalue late maturing players to counteract a perceived bias against early maturing players. Such behaviour would be an example of social desirability bias, where actions are influenced by a desire to conform to perceived social norms or expectations [36]. This bias is supported by the notion that coaches favoured players with a lower predicted adult height than average. This tendency might even be counterproductive, as successful adult players are generally taller than less successful players [37]. In addition, AB coaches selected players with a significantly lower %PAH than BB coaches and the average player, but when the ability to select relatively less mature players was severely restricted in BB, no such trend was observed. However, BB coaches still selected smaller players with a lower predicted adult height than the average player, suggesting an attempt at such overcompensation.

Coaches ultimately favoured players who performed better on the physical performance tests, suggesting that coaches were more likely to select high-performing players, regardless of maturity. Indeed, better performance on the YYIRT test, vertical jump height, and 30-m on-ice sprint time were significantly correlated with the number of selections (Fig 6), but again

were not related to maturity (Fig 2), suggesting that maturity did not influence these fitness advantages in the current cohort of players about to enter the accelerated phase of the growth spurt. Coaches also showed a clear tendency to select players in an offensive position, who generally matured later than defensive players. Whether this is related to a preference for the position itself (i.e. a scoring position) or a preference for the player characteristics exhibited by the offensive players in this study remains uncertain.

An alternative explanation for coaches' selection decisions related to maturity is the "underdog hypothesis", which proposes that late-maturing players must develop distinct skills to match the performance of their more mature teammates [38], or simply possess these skills at an early age in order to survive. The validity of this hypothesis is supported by the observation that the coaches selected older but physically later maturing players in the bio-band, where there was less variance in maturity. This pattern suggests that the competitive advantage that comes from being among the least mature players in a group with marginal maturity differences may be an important factor in later success after the growth spurt if one manages to stay on the team, which is supported by previous studies [18]. In contrast, early maturing athletes may not be exposed to such competitive scenarios in their regular age groups, thus missing out on these developmental experiences, which may be detrimental to long-term development, as previously mentioned.

## Practical applications

The influence of maturity in this cohort of young athletes, many of whom are at the beginning of their growth spurt, is complex. However, its importance in youth ice hockey academies is obvious. Given the parallels to bio-banding that has been explored in soccer, it seems reasonable to integrate bio-banding alongside age-banding in ice hockey, particularly in elite clubs where its impact can be maximized. The implementation of bio-banding can increase development opportunities for all players and ensure a more balanced maturity distribution. To make the most effective use of bio-banding, it would be beneficial to introduce it before the growth spurt accelerates and begins to significantly affect playing styles, positional roles, and selection decisions, as it can be challenging to influence positional and player roles later on. An early start may also allow younger players a smoother transition into this system before the inter-individual differences in size and physique begin to grow significantly. Although this was not tested in this study, it may therefore be advisable to introduce bio-banding at U11 or U12, as no players are likely to have reached their maximum growth rate at this age.

## Limitations

This is the first study to examine bio-banding in youth ice hockey players and the role of maturity status in elite male U13 and U14 players. In determining maturity status and timing, an updated reference population consisting of individuals with the same nationality was used, which likely improved the accuracy of the estimate. However, it should be noted that the Khamis Roche method, like any other non-invasive method for estimating maturity, has its limitations. It uses a prediction equation derived from samples of American adolescents of Northern European descent and applied to this Swedish cohort, where it has not been validated, with self-reported parental height. Furthermore, only two male age groups within one organization participated in the study, and the results should not necessarily be generalized to other age groups or female players. Nonetheless, the promising results can serve as preliminary evidence for initiatives to raise awareness of the role of biological maturation and to introduce bio-banding in ice hockey.

## Conclusions

This pilot study examined bio-banding in adolescent ice hockey players. The results showed that maturity status correlates with absolute strength, but not with speed or endurance, and that early-maturing players were more likely to play in a defensive position than late-maturing players. Bio-banding was perceived to create new challenges and a more individualized learning experience, emphasizing teamwork and without increasing the perceived risk of injury. High performance in physical fitness tests was important for selection by coaches, regardless of game format. Coaches favoured players with lower maturity status in age-banded games. Although bio-banding mitigated the importance of maturity, it emphasized chronological age, leading to a selection bias in favor of older players. Future studies should include a broader age range and focus specifically on the effects of bio-banding when fully implemented. They should also test selection bias in players throughout adolescence. In anticipation of future studies, ice hockey organizations should consider bio-banding as a complement to traditional age-banded activities.

## Supporting information

**S1 File. Supplementary data S1–S4 Figs and S1 Table.**
(PDF)

**S2 File. Raw data excluding birth date, height and weight.**
(XLSX)

## Acknowledgments

The authors would like to thank Christoffer Sundqvist from the Swedish Sports Confederation (Riksidrottsförbundet) and Bosön as well as Johan Krantz from Linköping Hockey Club for their valuable contributions.

## Author Contributions

**Conceptualization:** Oliver Lindholm, Erik Niklasson, John Lind, Tommy R. Lundberg.

**Data curation:** Oliver Lindholm.

**Formal analysis:** Oliver Lindholm, Tommy R. Lundberg.

**Investigation:** Oliver Lindholm, John Lind.

**Methodology:** Oliver Lindholm, Erik Niklasson, Daniele A. Cardinale, Tommy R. Lundberg.

**Project administration:** Tommy R. Lundberg.

**Resources:** John Lind, Daniele A. Cardinale.

**Supervision:** Erik Niklasson, Tommy R. Lundberg.

**Validation:** Oliver Lindholm.

**Visualization:** Oliver Lindholm.

**Writing – original draft:** Oliver Lindholm.

**Writing – review & editing:** Oliver Lindholm, Erik Niklasson, John Lind, Daniele A. Cardinale, Tommy R. Lundberg.

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
