## [Decision Letter · Decision Letter 0]

24 Jun 2024

PONE-D-24-20888A pilot study on bio-banding in male youth ice hockey: players' perceptions and coaches' selection preferencesPLOS ONE

Dear Dr. Lundberg,

Thank you for submitting your manuscript to PLOS ONE. After careful consideration, we feel that it has merit but does not fully meet PLOS ONE’s publication criteria as it currently stands. Therefore, we invite you to submit a revised version of the manuscript that addresses the points raised during the review process.

We look forward to receiving your revised manuscript.

Kind regards,

Ender Senel, PhD

Academic Editor

PLOS ONE

Journal Requirements:

   "TRL has received financial compensation from the Swedish Ice Hockey Association for consultancy work. JL is employed by the Swedish Ice Hockey Association. EN has received reimbursement of travel expenses from the Swedish Ice Hockey Association."

We note that one or more of the authors are employed by a commercial company:  Swedish Ice Hockey Association 

3. In this instance it seems there may be acceptable restrictions in place that prevent the public sharing of your minimal data. However, in line with our goal of ensuring long-term data availability to all interested researchers, PLOS’ Data Policy states that authors cannot be the sole named individuals responsible for ensuring data access (http://journals.plos.org/plosone/s/data-availability#loc-acceptable-data-sharing-methods).

Reviewers' comments:

Reviewer's Responses to Questions

**Comments to the Author**

1. Is the manuscript technically sound, and do the data support the conclusions?

Reviewer #1: Yes

Reviewer #2: Yes

2. Has the statistical analysis been performed appropriately and rigorously? 

Reviewer #1: Yes

Reviewer #2: Yes

3. Have the authors made all data underlying the findings in their manuscript fully available?

Reviewer #1: Yes

Reviewer #2: Yes

4. Is the manuscript presented in an intelligible fashion and written in standard English?

Reviewer #1: Yes

Reviewer #2: Yes

5. Review Comments to the Author

Reviewer #1: I would like to thank you for the opportunity to review this study, which is well-structured and of great value for the development of sports and youth training. Overall, the study is well-organized, involving significant logistics that underscore its importance. The only drawback I find is the lack of a greater difference in maturation, which would have allowed for a clearer distinction of the maturation influence. However, this is probably the scenario most frequently encountered in sports with these age groups. Therefore, I congratulate the authors on their work and offer a few suggestions that might be considered for improving the document.

Introduction

Third paragraph – Please consider including the effects of maturation on the physical demands pointed out, i.e., not also the performance changes but what leads to that (which changes in Central Nervous System, metabolic changes…)

Line 70 to 74 – which is in line with other sports. Please consider strengthened this idea, including some references from other sports.

Line 85 – in males

Line 86 – please consider including “in this pilot study”, otherwise it seems that you are refereeing to other study.

Methods

Including a subtopic of “intervention” seems that it is a longitudinal study. I suggest changing to something like “Test application” or other expression.

Player perceptions – which is the questionnaire used? Is validated?

Discussion

For a better understanding of the results, maybe this session could be divided in subtopics as players and coaches’ point of view, discussing the physical characteristics (all tests performed), maturation (AB and BB) and perception/selection.

Limitations

Although very implemented, the Khamis-Roche analysis was not validated in Sweden. Therefore, it should be included in the limitation of the study these issues, indicating that some errors could be included as a result of differences in anthropometric nationalities.

Reviewer #2: This is a novel and interesting study that explores a relatively new concept and strategy for the development of young athletes. The study is well designed and unique in nature and it is clear that the authors have a sound understanding of both growth, maturation, its assessment and the practice of bio-banding. So many papers that I read on this subject of growth and maturation are very ill informed, so it is refreshing to have the opportunity to read an informed paper. The methods of analysis are sound and the presentation of the results is excellent. I have some minor suggestions but in general think this is an excellent submission that should be prioritised for publication.

P2. L35. The R value should be r and there is no need to include the 0 just .57 as in r=.57

P3, L55. I am not sure prerequisites is the best word to use here. I assume you mean that these athletes possess substantial differences in size and athleticism and therefore may require different challenges in order to develop optimally?

P3, L61. Bio-banding is more of a strategy than a framework

P3, L65. Premier League requires capitalisation

P3, L70. More mature not matured…..may also say athletic than fit

P4, L96. I assume this was the biological parents who were assessed or self-reported height please clarify

P6, L124. I assume the %PAH was compared against sex specific means and sds for children of the same age and sex? Please clarify.

P9, L25. No 0 values required for p values i.e., .13 not 0.13

P10, L225. I really like the graphs in figure 4, well done.

P11, L236. The info in Figure 5 does not look like the items responses on the likert scale. I think you have made an error here.

P11, L242. I would assume the older players in the BB games would also be likely to be more delayed in maturity for their ages than the younger players in the same format. This would support the bio-banding concept that later developers are perceived more favorably in such contexts.

P12, L272. Or maybe that coaches select players of varying maturity timing to certain positions?

P14, L322. It is equally plausible that the late maturing have to possess these abilities to survive rather than develop them. In fact some of the evidence would suggest the superior abilities of late developing soccer players is more likely to result from selection than challenge. Likelihood is both contribute but selection is more important.

6. PLOS authors have the option to publish the peer review history of their article (what does this mean?). If published, this will include your full peer review and any attached files.

Reviewer #1: **Yes: **Ana Filipa Braga Barroso Campos Silva

Reviewer #2: No

---

## [Editor Report · Decision Letter 1]

30 Jul 2024

A pilot study on bio-banding in male youth ice hockey: players' perceptions and coaches' selection preferences

PONE-D-24-20888R1

Dear Dr. Lundberg,

We’re pleased to inform you that your manuscript has been judged scientifically suitable for publication and will be formally accepted for publication once it meets all outstanding technical requirements.

Kind regards,

Ender Senel, PhD

Academic Editor

PLOS ONE

---

## [Editor Report · Acceptance letter]

2 Aug 2024

PONE-D-24-20888R1 

PLOS ONE

Dear Dr. Lundberg, 

I'm pleased to inform you that your manuscript has been deemed suitable for publication in PLOS ONE. Congratulations! Your manuscript is now being handed over to our production team.

Kind regards, 

on behalf of

Dr. Ender Senel 

Academic Editor

PLOS ONE